# Analysis of ANRIL Isoforms and Key Genes in Patients with Severe Coronary Artery Disease

**DOI:** 10.3390/ijms242216127

**Published:** 2023-11-09

**Authors:** Francisco Rodríguez-Esparragón, Laura B. Torres-Mata, Sara E. Cazorla-Rivero, Jaime A. Serna Gómez, Jesús M. González Martín, Ángeles Cánovas-Molina, José A. Medina-Suárez, Ayose N. González-Hernández, Lidia Estupiñán-Quintana, María C. Bartolomé-Durán, José C. Rodríguez-Pérez, Bernardino Clavo Varas

**Affiliations:** 1Research Unit, Hospital Universitario de Gran Canaria Dr. Negrín, 35010 Las Palmas de Gran Canaria, Gran Canaria, Spain; lbtm1002@gmail.com (L.B.T.-M.); scazorla@ull.edu.es (S.E.C.-R.); jaimeserna@hotmail.com (J.A.S.G.); josu.estadistica@gmail.com (J.M.G.M.); canovaspi@hotmail.com (Á.C.-M.); jose.medina@ulpgc.es (J.A.M.-S.); angonherb@gobiernodecanarias.org (A.N.G.-H.); lestqui@gobiernodecanarias.org (L.E.-Q.); cbardur@gobiernodecanarias.org (M.C.B.-D.); 2Fundación Canaria Instituto de Investigación Sanitaria de Canarias (FIISC), Hospital Universitario de Gran Canaria Dr. Negrín, 35010 Las Palmas de Gran Canaria, Gran Canaria, Spain; 3Instituto Universitario de Enfermedades Tropicales y Salud Pública de Canarias de la Universidad de La Laguna, 38296 San Cristobal de La Laguna, Tenerife, Spain; 4CIBER de Enfermedades Infecciosas (CIBERINFEC), Instituto de Salud Carlos III, 28029 Madrid, Spain; 5Department of Specific Didactics, University of Las Palmas de Gran Canaria, 35004 Las Palmas de Gran Canaria, Gran Canaria, Spain; 6Department of Internal Medicine, University of La Laguna, 38200 La Laguna, Tenerife, Spain; 7Deparment of Cardiovascular Surgery, Hospital Universitario de Gran Canaria Dr. Negrín, 35010 Las Palmas de Gran Canaria, Gran Canaria, Spain; 8CIBER de Enfermedades Respiratorias, Instituto de Salud Carlos III, 28029 Madrid, Spain; 9Chronic Pain Unit, Hospital Universitario de Gran Canaria Dr. Negrín, 35010 Las Palmas de Gran Canaria, Gran Canaria, Spain; 10Deparment of Neurology and Clinical Neurophysiology, Hospital Universitario de Gran Canaria Dr. Negrín, 35010 Las Palmas de Gran Canaria, Gran Canaria, Spain; 11Vice Chancellor of Research, Universidad Fernando Pessoa Canarias, 35002 Santa María de Guía de Gran Canaria, Gran Canaria, Spain; jcrodriguez@ufpcanarias.es; 12Radiation Oncology Department, Hospital Universitario de Gran Canaria Dr. Negrín, 35010 Las Palmas de Gran Canaria, Gran Canaria, Spain; 13Universitary Institute for Research in Biomedicine and Health (iUIBS), Molecular and Translational Pharmacology Group, University of Las Palmas de Gran Canaria, 35016 Las Palmas de Gran Canaria, Gran Canaria, Spain; 14Spanish Group of Clinical Research in Radiation Oncology (GICOR), 28290 Madrid, Spain

**Keywords:** CDKN2B-AS1 (ANRIL), CDKN2A, VEGF, KDR, cardiovascular risk, coronary artery disease, outcome, gene expression

## Abstract

ANRIL (Antisense Noncoding RNA in the INK4 Locus), also named CDKN2B-AS1, is a long non-coding RNA with outstanding functions that regulates genes involved in atherosclerosis development. ANRIL genotypes and the expression of linear and circular isoforms have been associated with coronary artery disease (CAD). The CDKN2A and the CDKN2B genes at the CDKN2A/B locus encode the Cyclin-Dependent Kinase inhibitor protein (CDKI) p16INK4a and the p53 regulatory protein p14ARF, which are involved in cell cycle regulation, aging, senescence, and apoptosis. Abnormal ANRIL expression regulates vascular endothelial growth factor (VEGF) gene expression, and upregulated Vascular Endothelial Growth Factor (VEGF) promotes angiogenesis by activating the NF-κB signaling pathway. Here, we explored associations between determinations of the linear, circular, and linear-to-circular ANRIL gene expression ratio, CDKN2A, VEGF and its receptor kinase insert domain-containing receptor (KDR) and cardiovascular risk factors and all-cause mortality in high-risk coronary patients before they undergo coronary artery bypass grafting surgery (CABG). We found that the expression of ANRIL isoforms may help in the prediction of CAD outcomes. Linear isoforms were correlated with a worse cardiovascular risk profile while the expression of circular isoforms of ANRIL correlated with a decrease in oxidative stress. However, the determination of the linear versus circular ratio of ANRIL did not report additional information to that determined by the evaluation of individual isoforms. Although the expressions of the VEFG and KDR genes correlated with a decrease in oxidative stress, in binary logistic regression analysis it was observed that only the expression of linear isoforms of ANRIL and VEGF significantly contributed to the prediction of the number of surgical revascularizations.

## 1. Introduction

Cardiovascular diseases (CVDs) are the leading cause of global mortality and a major contributor to disability [1]. Coronary artery disease (CAD) is the most prevalent type of CVD [2]. Damage to the vascular endothelium is the first step in the cascade of events that lead to CAD.

The first series of genome-wide association studies (GWAS) for CAD identified a novel CAD risk locus on chromosome 9p21.3 [3,4,5,6]. To date, the 9p21.3 locus is the most robust and frequently replicated risk locus of CAD, among the numerous CAD risk loci identified by GWAS [7,8]. Chr9p21 risk is independent of classically known CAD risk determinants, such as dyslipidemia, diabetes mellitus, age, and sex [9]. However, the causative gene or genes for CAD at the 9p21.3 locus is at least partially unknown [10].

The CDKN2A/B locus at the 9p21.3 region encompasses three major tumor suppressor genes that are joined in a gene cluster: p16^INK4a^, p15^INK4b^, and p14^ARF^ [11]. The cyclin-dependent kinase inhibitor 2A (CDKN2A) gene, using a different reading frame, encodes the p16^INK4a^ and p14^ARF^ proteins. The CDKN2B gene encodes the p15^INK4b^. CDKN2B-AS1, known as antisense non-coding RNA in the INK4 locus (ANRIL), is transcribed from the opposite strand to the CDKN2B gene and encodes a long noncoding RNA (lncRNA) [12]. Single nucleotide polymorphisms (SNPs) that alter the expression of CDKN2B-AS are associated with CAD and with a human healthy life expectancy, as well as diabetes and cancers [12,13]. 

Multiple alternatives of ANRIL-processed transcripts, some of which may take the form of circular RNA molecules, have been detected [14,15,16]. The circular ANRIL isoforms originate from the linear ANRIL through a back-splicing mechanism [17]. The expression of certain linear and circular ANRIL transcripts shows tissue-specific patterns; however, the majority of studies have found that linear ANRIL isoforms significantly contribute to atherosclerosis development and CAD [12,13,15,18,19,20]. In contrast, circular ANRIL isoforms have been shown to confer an atheroprotective role [18,19,21,22,23].

Several molecular mechanisms have been described that explain how both linear and circular ANRIL isoforms might influence CAD and the progression of atherosclerosis. Thus, it has been observed that an altered ANRIL expression or dysfunction can influence the expression of the CDKN2A and CDKN2B genes [11]. The cyclins encoded by these genes play crucial roles in cell cycle regulation and apoptosis, subsequently influencing cell proliferation and senescence [24,25]. Further emphasizing its importance, altered ANRIL expression, especially in the context of modulating apoptosis and senescence, has been linked to an increased vulnerability of atherosclerotic plaques [12]. This is believed to be due to its effects on vascular smooth muscle cells and macrophages within these plaques [26]. Additionally, it has been described that ANRIL can recruit Polycomb repressive complexes (PRC1 and PRC2) to specific genomic sites [27]. These complexes trigger epigenetic modifications that suppress the transcription of numerous genes, some of which are involved in atherosclerotic development pathways [12,27]. Also, there is emerging evidence suggesting that ANRIL is involved in inflammatory pathways, potentially via modulating the NF-κB signaling cascade or influencing the activities of inflammatory cells, such as macrophages [28]. Moreover, there are described ANRIL interactions with specific microRNAs (miRNAs) that are pivotal in atherosclerosis. By acting as a “sponge” for these miRNAs, ANRIL may affect their regulatory capabilities on target genes, subsequently modulating atherosclerotic processes [29,30,31,32,33]. Finally, it has been shown that ANRIL regulates VEGF expression. VEGF is an important regulator of angiogenesis, lymphopoiesis and lymphangiogenesis, oxidative stress, lipid metabolism, and inflammation [34].

In this study, we analyzed the associations between the linear, circular, and linear-to-circular expression ratios of the ANRIL gene, in conjunction with CDKN2A, VEGF, and its kinase insert domain-containing receptor (KDR). Our focus was on evaluating their correlations with cardiovascular risk parameters and overall mortality rates in high-risk coronary patients who were slated for coronary artery bypass grafting surgery (CABG).

## 2. Results

One hundred sixty-three patients were recruited during the study period. Clinical, biochemical, and analyzed gene expression variables of the analyzed patients are depicted in Table 1. Table 2 shows the distribution of medical conditions and habits of evaluated patients, stratified by gender. 

### 2.1. ANRIL and CDKN2A Gene Expression and CAD Risk Factor

Males had higher linear and linear-to-circular ANRIL expression values than women. For ANRIL expression, a median value of 153 (IQR 33–754) was found for men and 84 (IQR 17–278) for women (*p* = 0.04). For linear-to-circular ANRIL expression, a median value of 2 (IQR 0.32–17) was found for men and 0.42 (IQR 0.48–5.7) for women (*p* = 0.01). No differences were observed for circular isoforms and CDKN2A expression values. 

There were statistically significant differences in circular ANRIL expression and the linear-to-circular expression ratio, according to hypertensive status. The expression of linear isoforms and the expression ratio between linear and circular isoforms of ANRIL were found to be different in hypertensive individuals compared to normotensive ones. The median value of linear isoforms of ANRIL in hypertensives was 123 (IQR 25–397) and 323 (IQR 67–757) in normotensives (*p* = 0.03). The median expression value of the ratio between linear and circular isoforms of ANRIL was 1.2 (IQR 0.2–7.5) in hypertensives and 7.4 (IQR 1.2–25) in normotensives (*p* = 0.05). There was a positive and significant correlation between linear ANRIL expression and BMI values (ρ = 0.224, *p* = 0.012, N = 125). Also, there was a positive and significant correlation between BMI values and linear-to-circular ANRIL expression values (ρ = 0.231, *p* = 0.04, N = 114) (Figure 1a,b). The expression levels of ANRIL linear isoforms differed according to a previously diagnosed of obesity. The median value was 748 (IQR 262–2742) in obese patients and 254 (IQR 58–897) in non-obese patients (*p*-value = 0.0057). When categorizing patients based on their mean IMC value, a significant difference was observed. Thus, the median linear ANRIL value was 107 (IQR 14–487) in patients with an IMC of 27 Kg/m^2^ or lower, while it was 166 (IQR 64–660) in those with an IMC higher than 27 Kg/m^2^ (*p*-value = 0.01023). No differences in the linear, circular, linear-to-circular ratio, and CDKN2A gene expression were found according to diabetic or dyslipidemic conditions, nor with respect to tobacco consumption. There were negative and significant correlations between circular ANRIL expression and CDKN2A gene expression and TBARS levels (rho = −0.2404738, S = 567281, *p*-value = 0.004211 and rho = −0.1871373, S = 628466, *p*-value = 0.02323, respectively). 

### 2.2. ANRIL and CDKN2A Gene Expression and CAD Severity

The linear, circular, and linear-to-circular ANRIL expression ratio and CDKN2A gene expression were analyzed, with respect to CAD severity. We found that, among patients stratified by the presence or absence of left ventricular dysfunction, there was a significant difference in the ANRIL linear-to-circular expression ratio. The median value was 0.74 (IQR 0.53–1) in patients with normal ventricular function, whereas a median value of 0.89 (IQR 0.6–1.18) was found in patients with ventricular dysfunction (*p* = 0.03). 

There was a significant correlation between the patient’s number of affected vessels and the number of surgical revascularizations that were performed (ρ = 0.525, *p* < 0.0001, N = 163). In Figure 2a–c, the boxplots illustrate the expression pattern of linear ANRIL isoforms, the expression ratio of linear-to-circular, and the expression of CDKN2A in accordance with the number of affected vessels. Figure 3a–c presents boxplots depicting the same parameters, but relative to the number of revascularizations. Interestingly, when dichotomized values of affected vessel numbers were analyzed, they bordered the threshold of significance in relation to the linear-to-circular expression ratio (W = 1335, *p* = 0.49) (Figure 2b). Nonetheless, when considering individual expressions of linear or circular ANRIL isoforms or CDKN2A gene expression, no variations were discerned regarding either vessel number. However, the linear ANRIL expression and the ANRIL expression ratio showed significant differences with respect to the number of surgical revascularizations (χ^2^ = 5, *p* = 0.025 and χ^2^ = 4.3, *p* = 0.036, respectively) (Figure 3b).

Furthermore, no differences in linear or circular ANRIL isoforms expression or CDKN2A gene expression were found when considering the presence or absence of aortic trunk lesion. 

Patients were divided into high- and low-expression categories based on medium and median values of linear and circular ANRIL and CDKN2A gene expression levels. Despite this segmentation, no association was observed between all-cause mortality and the analyzed gene expressions of ANRIL and CDKN2A.

### 2.3. Plasma VEGF, VEGF, and KDR Gene Expression and CAD Risk Factor

No differences were observed in VEGF and KDR gene expression according to sex. The median VEGF concentration was 31.12 (IQR 9.6–52), being significantly different between men and women (*p* = 0.049) with a median value for men of 37 (IQR 12.5–55), while for women it was 23 (IQR 7–32). However, no correlation was observed between VEGF expression and VEGF plasma levels in fully evaluated patients, also after stratifying for sex. VEGF and KDR gene expression were negatively correlated with TBARS levels (ρ = −0.253, *p* = 0.002, N = 147 and ρ = −0.202, *p* = 0.03, N = 116, respectively). The median VEGF concentration differed significantly between patients with and without a previous diagnosis of dyslipidemia (*p* = 0.016). For those with dyslipidemia, the median value was 37 (IQR 12.5–55), whereas for those without dyslipidemia at the time of diagnosis, the median was 19 (IQR 10–52). However, no differences in VEGF or KDR gene expression were observed with respect to diabetes and hypertension status, nor with respect to tobacco consumption.

### 2.4. Plasma VEGF, VEGF, and KDR Gene Expression and CAD Severity

VEGF gene expression correlated with the number of surgical revascularizations (ρ = 0.219, *p* = 0.007, N = 153) and with linear and circular ANRIL gene isoforms expression (ρ = 0.304, *p* < 0.001, N = 152 and ρ = 0.465, *p* < 0.0001, N = 141). Also, VEGF expression positively correlated with CDKN2A gene expression (ρ = 0.522, *p* < 0.001, N = 131). However, no associations were observed between VEGF and KDR gene expression levels with respect to evaluated CAD severity parameters. According to the medium and median values of both VEGF and KDR gene expression levels, all patients were divided into high- and low-expression groups. No association with mortality for all causes was observed for analyzed gene expressions.

### 2.5. Gene Expressions as Predictors of Surgical Revascularizations

The contribution of the evaluated gene expressions, as well as clinical, biochemical, and anthropometric variables, were analyzed using logistic binary models. As dependent variables, we evaluated the number of affected vessels, the number of grafting procedures, and all-cause mortality. 

Table 3, Table 4 and Table 5 shows the results, considering univariate models, a multivariate model, and the optimal model for each evaluated dependent variable.

No significant predictors were obtained either in single- or multivariate analysis of the number of affected vessels. On the contrary, a significant univariate contribution was obtained for the linear-to-circular ANRIL expression ratio and VEGF gene expression as predictors of the number of surgical revascularizations performed. Also, considering the CABG number as a dependent variable, the optimal multivariate model shows that linear ANRIL and VEFG gene expressions were significant predictors. 

Neither univariate nor multivariate models were obtained with significant predictors of all-cause mortality. Non-prospective assessment of gene expression does not appear to contribute significantly to all-cause mortality.

## 3. Discussion

ANRIL is subjected to a variety of splicing patterns producing multiple isoforms [12]. Differences in the biological effects, according to the differential expression of a particular ANRIL linear isoform and cell types, have been previously described [13,14,15,35,36,37]. Also, it has been observed that, in coronary arteries and peripheral blood mononuclear cells (PBMCs) from CAD patients, the expressed ANRIL linear isoforms result from specific exon combinations [13,14,15]. Increased ANRIL linear isoform expression increases, in turn, proatherogenic cell activities like proliferation and reduced apoptosis, as well as the differential expression of hundreds of genes, without affecting the expression of CDKN2A and CDKN2B genes [12,38]. In proliferative cells, linear ANRIL isoforms prevent cell senescence by repressing INK4 genes through the recruitment of Polycomb-group proteins [21]. 

Although there have been some discordant results [20,39], most linear isoforms have been associated with an increased risk of atherosclerosis and CAD severity [12,19,35,38]. In addition to the presence of risk SNPS that are also ANRIL gene transcription modifiers [29], conflictive results are thought to occur because ANRIL isoforms’ abundance and functions depend on the tissue and/or cell type [13,15] and on the existence of circular forms [19,31,37]. Circular ANRIL isoforms (circANRIL) are formed from linear isoforms by back-splicing [12,18,31] and, contrarily to linear ANRIL isoforms, circANRIL confer atheroprotection by regulating ribosomal RNA (rRNA) maturation [16,31] and microRNA sponging [23,32,33]. Accordingly, the atherogenic or atheroprotective roles of ANRIL seem to depend, among other factors, on the predominant expression of particular isoforms in different cell types and the balance of linear and circANRIL expression [14,23], as well as the metabolic control that they exert on genes, microRNAs, and proteins, which in turn are involved in cell proliferation or senescence.

VEGF is an important regulator of angiogenesis, lymphopoiesis and lymphangiogenesis, oxidative stress, lipid metabolism, and inflammation [40]. VEGF accelerates vascular injury by promoting endothelial cell migration, proliferation, and vascular permeability [40,41]. ANRIL expression is increased under inflammatory stimuli [28] and increased ANRIL levels regulate in turn VEGF expression, activating the NF-κB pathway, recruiting PCRC2 or p300 and regulating the miR-200b expression [34,38]. 

In the analyzed blood samples of patients with severe atherosclerosis collected before undergoing coronary artery bypass grafting (CABG), a positive and significant correlation between the values of the expression ratio of linear-to-circular isoforms of ANRIL and linear ANRIL isoforms was found. In partial agreement with previous observations, we found in univariate analysis that the linear-to-circular expression ratio and circANRIL isoforms differed with respect to hypertensive status. Accordingly, lower levels of ANRIL in hypertensive CAD patients, with respect to non-hypertensive CAD patients, have been observed [23]. Transcriptome studies have also discovered the contributions of ANRIL, AK098656, MEG3, H19, PAXIP1-AS1, TUG1, GAS5, CASC2, and CPS1-IT, among other long non-coding RNAs, to the pathophysiology of hypertension [42]. We also found a significant correlation between BMI and linear and circular ANRIL isoforms, and differences in linear and circANRIL, according to previously diagnosed obesity. As previously reported, there is a significant described role for several lncRNAs, including linear ANRIL isoforms, in the regulation of inflammatory pathways associated with obesity [43,44]. Also, univariate analysis showed that the ANRIL expression ratio was a predictor of the number of affected vessels and the CABG number, whereas VEGF gene expression was a predictor of the number of surgical revascularizations. 

We have evaluated an ANRIL amplicon that is in common with at least 17 of the 28 ANRIL linear transcripts and therefore represents the most frequent linear isoforms found in PBMCs from CAD patients. Additionally, the cirANRIL amplicon tested represents the most frequent circular ANRIL form described. Disease severity in CAD patients was evaluated by testing for the significant contributions of genes and clinical, biochemical, and anthropometric variables to the number of affected vessels, the number of CABGs, and all-cause mortality. We observed that the number of surgical revascularizations performed with internal mammary grafts was significantly correlated with the number of affected vessels in the patients evaluated. However, the joint analysis of gene expression and the different variables analyzed resulted in better and more precise multivariate models in the classification of the number of revascularizations than those obtained for the number of vessels or in the evaluation of mortality for all causes. Moreover, in the logistic binary regression analysis, both linear ANRIL and VEGF gene expression were predictors of the number of surgical revascularization procedures. However, no univariate or multivariate logistic models were obtained in which genes were significant predictors of affected vessels and all-cause mortality. 

Nevertheless, these results should be considered with caution, considering our study limitations. Among others, this was a small sample study without a control group. A long-term assessment of overall survival would have been more informative than the assessment of all-cause mortality. Moreover, some ANRIL isoforms are predominant in specific cell types that are relevant in the atherosclerosis process, such as endothelial cells. Thus, the use only of peripheral blood mononuclear cells to determine the expression of the evaluated genes constitutes another significant limitation of the study. Furthermore, the prospective evaluation of gene expression would probably have also provided information of interest. 

In conclusion, this study suggests that the assessment of the expression of linear and circular isoforms of ANRIL expression and VEGF are useful predictors of the CAGB number and CAD severity. However, linear ANRIL expression and the linear-to-circular expression ratio were found to be highly correlated, suggesting that a single evaluation of the ANRIL expression ratio does not provide more information than that obtained from the sole determination of the linear expression of ANRIL. 

## 4. Materials and Methods

### 4.1. Patients

Blood samples were collected from 163 men and women undergoing coronary artery bypass grafting (CABG) at Hospital Universitario de Gran Canaria Dr. Negrín (HUGCDN), between April 2017 and April 2022. CABG indication was based on standard clinical and angiographic criteria. The study approval was granted by the Ethics Committee of the HUGCDN (reference 140157), and informed consent was obtained from all individual donors in accordance with Spanish legislation. Research was carried out in compliance with the Helsinki Declaration (http://www.wma.net/e/policy/b3.htm, accessed on 1 September 2023). CAD severity data were collected from medical records at the end of the study period. In that temporary evaluation it was observed that ten patients were exitus.

### 4.2. Biochemical Determinations

The reactive oxygen species bypass product malondialdehyde was measured in serum samples by thiobarbituric acid reactive substances (TBARS) assay. Serum VEGF levels were measured by an ELISA assay, as described by the manufacturer (Thermo Fisher, Waltham, MA, USA). 

### 4.3. Gene Expression 

Total RNAs were isolated from patients’ blood samples using Trizol reagent (Thermo Fisher Scientific, Madrid, Spain), according to the manufacturer’s instructions. cDNA synthesis was performed using the iScript™ kit (Biorad, Hercules, CA, USA). Analysis of the relative gene expression of CDKN2B-AS1 (ANRIL) was performed, using previously described primers with the sequences: 5′-TCACTGTTAGGTGTGCTGGAAT-3′ and 5′-CCTCTGATGGTTTCTTTGGAGT-3′. These primers amplified exon 6 of the actual annotated ANRIL exons structure. Circular ANRIL was amplified with previously described primers [11]. These primers amplified a fragment of the predominant circANRIL5–7 isoform, which consisted of exons 5, 6 and 7, where exon 7 was non-canonically spliced to exon 5. A 150bp fragment of the VEGFA gene was amplified with the primers 5′-TGGGCCTTGCTCAGAGCGGA-3′ and 5′-GCTCACCGCCTCGGCTTGTC-3′. The primer sequences for CDKN2A and KDR amplifications were as previously described [45,46]. 

### 4.4. Statistical Analysis

Data were presented as means ± standard deviations (SD) for continuous variables and as frequencies and percentages for categorical variables. The normality of continuous variables was assessed using a Kolmogorov–Smirnov test. Correlations and group comparisons were conducted using parametric or non-parametric tests, based on the normality test results. Analyzed variables included hypertension, diabetes, dyslipidemia, and obesity. Tobacco consumption was categorized as current smokers, including ex-smokers who had quit within one year, and non-smokers. Ventricular function was categorized as normal or as mild/moderate/severe dysfunction. The presence or absence of an aortic trunk lesion and the number of surgical revascularization procedures were recorded. The number of affected vessels was coded as one or two versus three or more. The grafting revascularization number was coded as one affected vessel or more than one vessel. Gene expression comparisons based on categorical values used the Wilcoxon test for dummy variables or the Kruskal–Wallis test for categorical variables with more than two categories. The multivariable association between specific coronary risk factors and evaluated gene expression was assessed using binary logistic regression. To identify independent predictors of the number of affected vessels, the number of CABG procedures, and all-causes mortality, stepwise logistic regression analyses were performed, computing both forward stepwise and backward selection to choose an optimal simple model without compromising accuracy. A value of *p* < 0.05 was considered statistically significant.

## Figures and Tables

**Figure 1 ijms-24-16127-f001:**
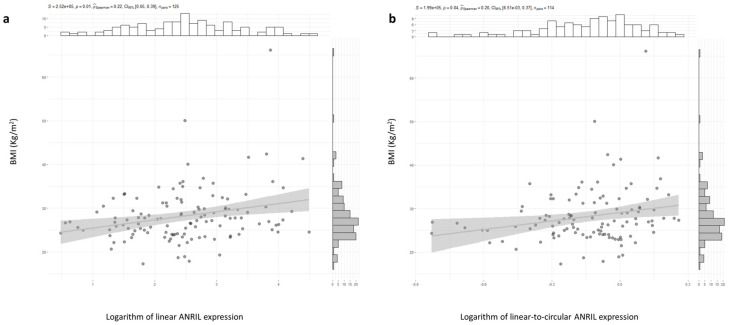
(**a**) Linear ANRIL expression and (**b**) linear-to-circular ANRIL expression ratio correlated with BMI in CAD patients. Each position of a point on the graph is determined by the values of both variables for that observation. The points with darker colors occur when observations overlap.

**Figure 2 ijms-24-16127-f002:**
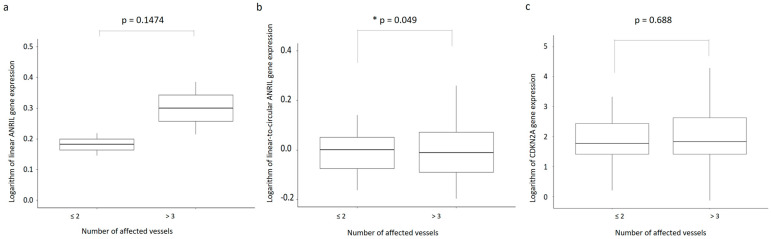
Boxplots of the (**a**) logarithm of the ANRIL linear expression, (**b**) logarithm of linear-to-circular ANRIL expression ratio, and **(c)** logarithm of the CDKN2A gene expression according to the number of affected vessels. * *p* < 0.005.

**Figure 3 ijms-24-16127-f003:**
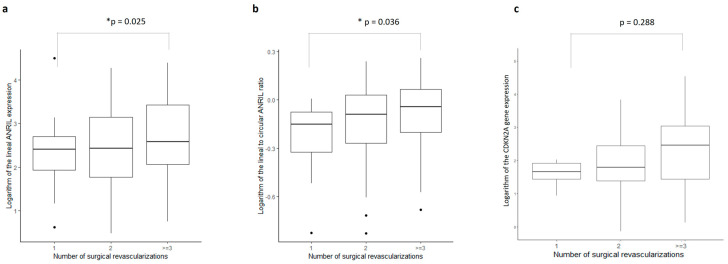
Boxplots of the (**a**) logarithm of the ANRIL linear expression, (**b**) logarithm of linear-to-circular ANRIL expression ratio, and (**c**) logarithm of the CDKN2A gene expression according to the number of revascularization procedures. * *p* < 0.05.

**Table 1 ijms-24-16127-t001:** Baseline characteristics of CAD patients.

	N.	Mean	St. Dev.	Min.	Max.
Age (years)	163	70.963	9.438	35	101
BMI (Kg/m^2^)	135	28.206	6.115	17.290	66.120
SBP (mmHg)	161	129.012	21.429	85	200
DBP (mmHg)	161	72.453	14.312	48	169
Cholesterol (mg/dL)	141	139.553	41.881	59	355
HDL cholesterol (mg/dL)	118	40.869	13.914	17.000	138.000
LDL cholesterol (mg/dL)	104	77.618	28.203	22.900	151.000
Triglycerides (mg/dL)	141	145.352	71.788	55.000	585.000
TBARS (µM)	153	2.285	1.182	1.118	7.302
Plasma VEGF (pg/mL)	40	40.951	45.445	4.432	228.102
Glycemia (mg/dL)	162	143.296	57.386	67	442
Urea (mg/dL)	148	54.209	36.086	13	258
Urate (mg/dL)	54	5.244	1.857	1.180	11.140
Creatinine (mg/dL)	161	1.232	0.931	0.500	8.400
GFR (CKD-EPI) (ml/min^−1^)	161	70.177	26.402	5.480	118.660
GFR (MDRD-IMDS) (ml/min^−1^)	160	72.657	30.549	6.200	166.880
Folic acid (ng/mL)	23	6.996	4.381	1.800	18.800
Vitamin B_12_ (pg/mL)	29	357.828	197.455	159	1021
Leukocytes (×10^3^/µL)	162	10.074	4.789	1.990	31.710
Neutrophils (×10^3^/µL)	162	7.214	4.732	1.260	30.310
Lymphocytes (×10^3^/µL)	162	1.832	0.884	0.240	5.080
Monocytes (×10^3^/µL)	162	0.777	0.394	0.030	3.200
Eosinophils (×10^3^/µL)	162	0.205	0.192	0.000	0.950
Basophils (×10^3^/µL)	162	0.043	0.031	0.000	0.180
Red blood cells (×10^3^/µL)	162	4.000	0.816	2.440	6.060

BMI: body mass index; SBP: systolic blood pressure; DBP: diastolic blood pressure; HDL cholesterol: high-density lipoprotein cholesterol; TBARS: thiobarbituric acid reactive substances; plasma VEGF: plasma vascular endothelial growth factor; GRF (CKD-EPI): glomerular filtration rate, chronic kidney disease epidemiology collaboration; GFR (MDRD-IMDS): glomerular filtration rate, modification of diet in renal disease study group equation.

**Table 2 ijms-24-16127-t002:** Medical conditions and habits of patients by gender.

Condition/Habit	Total (%)	Male (%)	Female (%)	*p* Value
Number of Patients	163	131	32	-
Hypertensive patients	82.8	81.6	87.5	0.434
Diabetic patients	60.7	57.2	75	0.065
Patients with dyslipidemia	79.7	77.8	87.5	0.224
Smokers	47.2	54.2	18.8	<0.001
Patients with overweight	15.9	15.2	18.7	0.630
Patients with LVD *	58.3	55.7	68.7	0.180
Patients with aortic trunk lesion	33	34.6	28.1	0.485
Patients with ≥2 affected vessels	92	93.9	84.4	0.075
Patients with ≥2 revascularizations	82.8	87	65.6	0.004
All causes of death	26.4	26.7	25	0.843

* LVD: left ventricular dysfunction.

**Table 3 ijms-24-16127-t003:** Univariate and multivariate analysis of predictors of the number of affected vessels in CAD patients.

Variables	Univariate						Multivariate					Optimal Multivariate Model				
	N	Beta	SE	OR	CI 95	*p*-Value	Beta	SE	OR	CI 95	*p*-Value	Beta	SE	OR	CI 95	*p*-Value
Intercept							2.96	2.49	19.318	0.157–3574.719	0.235	2.83	1.1	16.926	2.91–328.83	0.01
Female sex	77	−0.58	0.59	0.56	0.18–1.86	0.325	−0.9	0.8	0.405	0.08–1.958	0.26	-	-	-	-	-
Hypertension	77	−1.51	1.08	0.22	0.01–1.25	0.161	−1.49	1.19	0.226	0.011–1.651	0.211	−1.57	1.09	0.209	0.011–1.21	0.15
Diabetes	77	0.46	0.52	1.59	0.57–4.49	0.375	0.72	0.64	2.063	0.599–7.657	0.258	-	-	-	-	-
Dyslipidemia	77	−0.17	0.64	0.85	0.21–2.81	0.795	0.21	0.81	1.239	0.234–5.99	0.791	-	-	-	-	-
Smoker	77	0.17	0.52	1.18	0.42–3.35	0.751	0.2	0.75	1.226	0.28–5.506	0.785	-	-	-	-	-
LVD	77	−0.45	0.52	0.64	0.22–1.78	0.392	−0.53	0.61	0.59	0.174–1.932	0.385	-	-	-	-	-
Aortic trunk	77	−0.82	0.53	0.44	0.15–1.24	0.122	−1.3	0.66	0.274	0.069–0.965	0.051	−0.85	0.54	0.426	0.14–1.21	0.113
BMI	77	0.01	0.04	1.01	0.93–1.1	0.857	−0.03	0.05	0.972	0.884–1.083	0.57	-	-	-	-	-
Lineal ANRIL	77	0.32	0.35	1.38	0.7–2.81	0.36	0.68	0.58	1.978	0.66–6.681	0.239	-	-	-	-	-
Circular ANRIL	77	−0.14	0.31	0.87	0.48–1.66	0.646	−0.17	0.38	0.841	0.401–1.798	0.645	-	-	-	-	-
Ratio ANRIL	77	1.26	1.06	3.53	0.15–86.8	0.43	-	-	-	-	-	-	-	-	-	-
CDKN2A	77	0.01	0.32	1.01	0.53–1.93	0.983	−0.56	0.58	0.571	0.169–1.752	0.335	-	-	-	-	-
VEGF	77	0.04	0.29	1.04	0.59–1.9	0.89	0.11	0.41	1.112	0.499–2.563	0.794	-	-	-	-	-
KDR	77	0.18	0.19	1.19	0.81–1.74	0.357	0.11	0.22	1.115	0.719–1.718	0.618	-	-	-	-	-
AUC ROC						0.7509					0.643					

Factors predicting the number of affected vessels were analyzed using univariate and multivariate logistic regression. The reference categories are as follows: for hypertension, diabetes, and dyslipidemia, it was ‘previously diagnosed disease’; for ‘smoker’, it was ‘current smoker’; for LVD, it was ‘mild to severe dysfunction’; and for ‘aortic lesion’, it was ‘presence of aortic trunk lesion’. LVD: left ventricular dysfunction; BMI: body mass index; CDKN2A: cyclin dependent kinase inhibitor 2A; VEGF: vascular endothelial growth factor; KDR: kinase insert domain receptor; AUC ROC: area under the ROC curve.

**Table 4 ijms-24-16127-t004:** Univariate and multivariate analysis of predictors of the number of surgical revascularizations in CAD patients.

Variables	Univariate						Multivariate					Optimal Multivariate Model				
	N	Beta	SE	OR	CI 95	*p*-Value	Beta	SE	OR	CI 95	*p*-Value	Beta	SE	OR	CI 95	*p*-Value
Intercept	-	-	-	-	-	-	−2.38	3.16	0.092	0–36.039	0.451	−1.41	2.37	0.245	0.001–19.76	0.554
Femalesex	77	−1	0.65	0.37	0.1–1.4	0.128	−1.59	1.28	0.204	0.013–2.241	0.212	−1.85	0.99	0.157	0.018–0.988	0.062
Hypertension	77	−0.91	1.09	0.4	0.02–2.38	0.403	0.03	1.61	1.026	0.025–22.30	0.987	-	-	-	-	-
Diabetes	77	0.6	0.61	1.82	0.54–6.26	0.328	2.07	1.12	7.953	1.17–109.03	0.063	1.77	0.87	5.851	1.185–39.824	0.043
Dyslipidemia	77	0.07	0.72	1.07	0.22–4.09	0.924	1.9	1.21	6.662	0.689–93.45	0.116	1.93	1.12	6.876	0.809–75.708	0.085
Smoker	77	0.09	0.61	1.1	0.33–3.75	0.881	0.1	1.14	1.11	0.122–12.43	0.927	-	-	-	-	-
LVD	77	0.41	0.62	1.5	0.45–5.44	0.513	−0.19	0.94	0.826	0.115–5.23	0.838	-	-	-	-	-
Aortictrunk	77	0.95	0.7	2.59	0.71–12.38	0.176	0.99	0.98	2.699	0.433–22.51	0.31	-	-	-	-	-
BMI	77	−0.01	0.04	0.99	0.91–1.09	0.768	−0.14	0.08	0.87	0.724–1.02	0.092	−0.14	0.07	0.872	0.75–1.009	0.06
LinearANRIL	77	0.82	0.45	2.27	0.99–5.86	0.067	1.69	0.88	5.419	1.184–42.45	0.055	1.57	0.72	4.8	1.314–23.896	0.029
CircularANRIL	77	−0.29	0.34	0.75	0.39–1.54	0.396	−0.99	0.63	0.371	0.087–1.1	0.118	−0.85	0.5	0.429	0.14–1.07	0.09
RatioANRIL	77	3.28	1.93	26.54	0.66–1436.8	0.09	-	-	-	-	-	-	-	-	-	-
CDKN2A	77	0.28	0.39	1.32	0.62–2.96	0.481	−1.26	0.8	0.284	0.046–1.182	0.118	−0.92	0.63	0.399	0.105–1.318	0.143
VEGF	77	0.85	0.43	2.33	1.07–5.85	0.048	2.73	1.06	15.369	2.65–183.23	0.01	2.35	0.84	10.516	2.378–67.635	0.005
KDR	77	0.3	0.22	1.34	0.87–2.06	0.172	0.38	0.32	1.459	0.783–2.85	0.237	-	-	-	-	-
AUCROC	-						0.887					0.863				

Factors predicting the number of surgical revascularizations were analyzed using univariate and multivariate logistic regression. The reference categories are as follows: for hypertension, diabetes, and dyslipidemia, it was ‘previously diagnosed disease’; for ‘smoker’, it was ‘current smoker’; for LVD, it was ‘mild to severe dysfunction’; and for ‘aortic lesion’, it was ‘presence of aortic trunk lesion’. LVD: left ventricular dysfunction; BMI: body mass index; CDKN2A: cyclin dependent kinase inhibitor 2A; VEGF: vascular endothelial growth factor; KDR: kinase insert domain receptor; AUC ROC: area under the ROC curve.

**Table 5 ijms-24-16127-t005:** Univariate and multivariate analysis of predictors of all causes of mortality in CAD patients.

Variables	Univariate						Multivariate					Optimal Multivariate Model				
	N	Beta	SE	OR	CI 95	*p*-Value	Beta	SE	OR	CI 95	*p*-Value	Beta	SE	OR	CI 95	*p*-Value
Intercept	-	-	-	-	-	-	2.77	2.94	16.019	0.064–7938	0.346	4.1	2.04	60.087	1.399–4712	0.045
Femalesex	77	1	0.61	2.73	0.79–9.08	0.103	1.39	0.94	4.017	0.654–28.5	0.14	1.16	0.72	3.18	0.772–13.6	0.108
Hypertension	77	−0.29	0.74	0.75	0.19–3.73	0.696	−0.04	0.97	0.958	0.151–7.521	0.965	-	-	-	-	-
Diabetes	77	−0.11	0.57	0.89	0.29–2.8	0.842	0.3	0.72	1.349	0.339–5.932	0.676	-	-	-	-	-
Dyslipidemia	77	−1.38	0.61	0.25	0.07–0.85	0.024	−1.35	0.87	0.261	0.044–1.437	0.122	−1.49	0.72	0.225	0.052–0.91	0.038
Smoker	77	−0.22	0.57	0.8	0.26–2.43	0.699	0.76	0.85	2.129	0.414–12.501	0.375	-	-	-	-	-
LVD	77	0.48	0.57	1.62	0.54–5.07	0.395	0.52	0.71	1.68	0.418–7.226	0.467	-	-	-	-	-
Aortictrunk	77	−0.87	0.63	0.42	0.11–1.36	0.17	−0.75	0.75	0.474	0.097–1.99	0.321	-	-	-	-	-
IMC	77	−0.21	0.08	0.81	0.69–0.93	0.007	−0.18	0.08	0.835	0.692–0.968	0.031	−0.18	0.08	0.838	0.708–0.96	0.022
LinealANRIL	77	−0.35	0.38	0.71	0.33–1.47	0.363	−0.46	0.73	0.63	0.141–2.513	0.527	-	-	-	-	-
CircularANRIL	77	0.09	0.34	1.09	0.53–2.05	0.797	0.1	0.43	1.105	0.451–2.569	0.818	-	-	-	-	-
RatioANRIL	77	−1.93	1.74	0.15	0–4.31	0.267	-	-	-	-	-	-	-	-	-	-
CDKN2A	77	0.08	0.35	1.09	0.54–2.16	0.813	0.72	0.78	2.047	0.46–9.931	0.356	-	-	-	-	-
VEGF	77	0.35	0.31	1.42	0.77–2.63	0.261	0.19	0.52	1.211	0.424–3.453	0.714	-	-	-	-	-
KDR	77	−0.13	0.21	0.88	0.59–1.34	0.537	−0.13	0.26	0.875	0.53–1.496	0.606	-	-	-	-	-
AUCROC	-						0.8135					0.7684				

Factors predicting mortality were analyzed using univariate and multivariate logistic regression. The reference categories are as follows: for hypertension, diabetes, and dyslipidemia, it was ‘previously diagnosed disease’; for ‘smoker’, it was ‘current smoker’; for LVD, it was ‘mild to severe dysfunction’; and for ‘aortic lesion’, it was ‘presence of aortic trunk lesion’. LVD: left ventricular dysfunction; BMI: body mass index; CDKN2A: cyclin dependent kinase inhibitor 2A; VEGF: vascular endothelial growth factor; KDR: kinase insert domain receptor; AUC ROC: area under the ROC curve.

## Data Availability

All data needed to evaluate the conclusions of the paper are present and tabulated in the main text. This article is the result of an original analysis of data from patients with coronary artery disease. The corresponding author has full access to all data in the study and has the final responsibility for the integrity of the data, the accuracy of the data analysis, and the decision to submit for publication. All data associated with the article are available if required in Excel or R, and as pictures in .tif or .jpg formats.

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
