# Peer review of "Analysis of ANRIL Isoforms and Key Genes in Patients with Severe Coronary Artery Disease"

_ijms, 2023, doi:10.3390/ijms242216127_

Round 1

Reviewer 1 Report

Comments and Suggestions for Authors

The article is devoted to the study of gene expression of lineal and circular ANRIL isoforms, CDKN2A, VEGF and KDR as factors that potentially could modify the progression of atherosclerosis. PBMCs of patients scheduled for CABG procedure were studied and serum concentration of VEGF, as well as the level of oxidative stress, was evaluated. The study is scientifically sound, but there are several concerns that have to be addressed:

1.       The title is vague and does not provide the essence of the article, being overloaded with abbreviations at the same time. It is better to formulate the concise main idea of the research and use it as a title.

2.       Table 1 – deciphering of the abbreviations below the table is required. Sex should be included in the baseline characteristics of patients, especially with the sex-dependent analysis of gene expression being provided below. The units for BMI, SBP and DBP are missing

3.       Lines 89 – 96 the results and not only p-level should be presented either in text or in figure/table.

4.       Lines 119-125 also, the presentation of the results is missing. The same is applicable to the whole 2.3, 2.4 sections – the results are represented neither in text, nor in illustrative material.

5.       Line 144 typo in VEGF.

6.       Tables 2-4 – deciphering of abbreviations is required

7.       Line 208 : In the analyzed samples of patients with severe atherosclerosis.. it is better to use: In the analyzed blood samples of patients with severe atherosclerosis…

8.       Lines 231-233: “We observed that the number of surgical revascularizations performed with internal mammary grafts was significantly correlated with the number of affected vessels in the patients evaluated.” – these results were not presented in the results section.

9.       The authors discuss that expression of ANRIL isoforms depends on the cell type. Meanwhile, only PBMCs were used for the analysis. This does not allow to make a certain conclusion regarding ANRIL expression in endothelial cells, which would have been of the greatest importance in atherogenesis. This should be mentioned among the limitations of the study.

Author Response

The authors would like to express their gratitude to the reviewer for their effort in reading and critically revising the manuscript, for their kind words, and for his timely comments. Based on your suggestions, we have proceeded to make the following changes:

1.- Certainly, the title includes numerous gene name abbreviations that could distort the research idea. In an attempt not to excessively modify the title while maintaining the evaluation of ANRIL isoforms as our main objective and in accordance with the reviewer's suggestion, we have decided to shorten the title. Thus, the title has been changed to “Analysis of ANRIL Isoforms and Key Genes in Patients with Severe Coronary Artery Disease”

2.- According to the reviewer's suggestions, information regarding the sex of the patients has been included. Thus, in the material and methods section, in the patients subsection, the percentage of men (80.4%) is now included in parentheses. There are no missing values of this variable so the difference corresponds to the percentage of recruited women. Likewise, the units of measurement are now reflected in the table 1. We also include at the bottom of the table the deciphering of the abbreviations.

3.- The gene distribution we analyzed did not follow a normal distribution. Consequently, we employed non-parametric tests to assess differences. Accordingly, results in lines 89-96 are presented as medians with accompanying percentiles, along with the p-values for each contrast.

4.- Accordingly to the reviewer´s suggestion, we have restructured the information data on the expression values of linear isoforms of ANRIL, the expression ratio between linear and circular isoforms, and the expression values of CDKN2A. This information is now shown in a revised Figure 2 and in a new Figure 3, presented as boxplots. Sections 2.3 and 2.4 of the manuscript were modified to include median values and interquartile ranges (IQRs) about differences evaluated on plasma VEGF levels. Previous content in the manuscript has been crossed out, and the new changes are highlighted in yellow.

5.- As suggested we make the appropriate correction.

6.- As suggested we make the appropriate corrections at the bottom of the corresponding tables.

7.- We corrected the sentence according to the reviewer's suggestion.

8.- Certainly, the correlation between the number of affected vessels and surgical revascularization procedures was discussed without being adequately presented in the manuscript. This correlation is now shown in section 2.2 of the text.

9.- As suggested by the reviewer, we include the following statement among the limitations of the study: “Furthermore, some ANRIL isoforms are predominant in specific cell types that are relevant in the atherosclerosis process, such as endothelial cells. Thus, the use only of peripheral blood mononuclear cells to determine the expression of the evaluated genes constitutes another significant limitation of the study.”

Reviewer 2 Report

Comments and Suggestions for Authors

In this manuscript, most of the results are not provided with figures, and scientific questions about why the authors conducted these measurements were also missing in the introduction part. For clinical studies, the information of patients should also be clearly presented, such as the gender, and diabetic percentage et al.

The manuscript tries to evaluate the association of linear and circular ANRIL with CAD risk profiles. Overall, the research is poorly designed, and the results are badly presented in the current version of the manuscript.

-When designing the clinical experiments in this manuscript, the critical information of the patients is missing, such as the gender ratio and diabetic percentage in the patients et al. Moreover, the N number for each parameter in Table 1 is totally different and the N number for Table 2-4 is fixed at 77. What are the criteria to include or exclude the samples?

-In the result part, most of the results are only described in the text without providing the data in the figures. This is not the way to present scientific data.

-The introduction part is not sufficient. The molecular mechanisms for the functions of linear and circular ANRIL in atherosclerosis should be clearly introduced, and what are the scientific questions the author wanted to answer? Why did the author pick ANRIL?

-For all the tables, the abbreviations should be clearly defined in the legends.

Author Response

We sincerely thank you for your valuable comments and suggestions. We have taken each of your observations seriously and have made improvements to our manuscript accordingly:

Results and Figures: We have revisited our manuscript and have incorporated an additional figure that more clearly illustrates our study's main findings. We believe these visualizations will offer readers an immediate and deeper understanding of the data presented.

Scientific Justification in the Introduction: We acknowledge that the initial introduction did not provide sufficient justification for our measurements. We have enriched this section to better explain the scientific context and relevance of the measurements we undertook, as well as the research questions we aimed to answer.

Patient Information: In line with your observation, we have added a detailed section presenting the demographic and clinical information of the patients included in our study. This includes gender, percentage of diabetic patients, and other relevant characteristics that might influence the interpretation of our findings.

To facilitate your review, we have crossed out the previous text and highlighted the changes made in yellow.

Differences in sample sizes between Tables: Your observation of the discrepancies in sample sizes across tables is valid. The reduced sample size in our multivariate analyses (N=77) compared to the initial univariate analysis arises from the necessity of complete data for each sample across all examined variables. Missing data on any parameter would naturally exclude that specific sample from the multivariate analyses. However, despite this reduced sample size in the multivariate analyses, we have taken rigorous steps to ensure the robustness and fit of our models. Particularly, we have utilized the Hosmer-Lemeshow goodness-of-fit test and the -2 log likelihood (-2LL) to evaluate our models. While we acknowledge the critiques of the Hosmer-Lemeshow test seeking "non-significance," it provides valuable insight when used judiciously. Additionally, a smaller -2LL indicates a better fit of the model to the data, further strengthening our confidence in our results. Our central aim was to discern if the evaluated genes' expression, when combined with other clinical variables, offers utility in assessing the number of revascularizations, the number of affected vessels, and all-cause mortality. The methodological checks and balances we have employed, even with our study's inherent limitations, vouch for the reliability and robustness of our conclusions.

We have worked diligently to ensure these points are more explicitly articulated in our revised manuscript for better clarity and context.

We thank you for your meticulous attention and hope you find these revisions satisfactory. We remain open to further suggestions and feedback to further improve our work.

Round 2

Reviewer 2 Report

Comments and Suggestions for Authors

The revised version has improved substantially compared with the original manuscript. Thanks for answering all my questions. I don't have more questions.